# Emerging Role of Noncoding RNAs in EGFR TKI-Resistant Lung Cancer

**DOI:** 10.3390/cancers14184423

**Published:** 2022-09-12

**Authors:** Jingwei Li, Peiyi Li, Jun Shao, Shufan Liang, Yuntian Wan, Qiran Zhang, Changshu Li, Yalun Li, Chengdi Wang

**Affiliations:** 1Department of Respiratory and Critical Care Medicine, Med-X Center for Manufacturing, Frontiers Science Center for Disease-Related Molecular Network, West China Medical School/West China Hospital, Sichuan University, Chengdu 610041, China; 2Department of Anesthesiology, West China Medical School/West China Hospital, Sichuan University, Chengdu 610041, China; 3West China Medical School/West China Hospital, Sichuan University, Chengdu 610041, China

**Keywords:** EGFR-TKI resistance, ncRNAs, lung cancer, mechanism

## Abstract

**Simple Summary:**

The introduction of epidermal growth factor receptor (EGFR)-tyrosine kinase inhibitors (TKIs) has revolutionized the treatment of lung cancer. Nevertheless, TKI resistance impedes therapeutic efficacy and its underlying mechanisms remain unclear. This review summarizes the potential mechanisms of noncoding RNAs (ncRNAs) in EGFR TKI-resistant lung cancer and their clinical applications. Moreover, we highlight the bottlenecks that urgently need to be addressed to promote the clinical application of ncRNAs.

**Abstract:**

Lung cancer accounts for the majority of malignancy-related mortalities worldwide. The introduction of epidermal growth factor receptor (EGFR)-tyrosine kinase inhibitors (TKIs) has revolutionized the treatment and significantly improved the overall survival (OS) of lung cancer. Nevertheless, almost all *EGFR*-mutant patients invariably acquire TKI resistance. Accumulating evidence has indicated that noncoding RNAs (ncRNAs), such as microRNAs (miRNAs), long noncoding RNAs (lncRNAs) and circular RNAs (circRNAs), have a central role in the tumorigenesis and progression of lung cancer by regulating crucial signaling pathways, providing a new approach for exploring the underlying mechanisms of EGFR-TKI resistance. Therefore, this review comprehensively describes the dysregulation of ncRNAs in EGFR TKI-resistant lung cancer and its underlying mechanisms. We also underscore the clinical application of ncRNAs as prognostic, predictive and therapeutic biomarkers for EGFR TKI-resistant lung cancer. Furthermore, the barriers that need to be overcome to translate the basic findings of ncRNAs into clinical practice are discussed.

## 1. Introduction

Lung cancer, the foremost malignancy in terms of mortality rate, can be divided into non-small cell lung cancer (NSCLC) and small cell lung cancer (SCLC) [1,2]. NSCLC, accounting for approximately 85% of lung cancers, is mainly composed of lung squamous cell carcinoma (LUSC), lung adenocarcinoma (LUAD) and large cell lung cancer [2]. Over the last few decades, the rapid development of therapeutic regimens such as targeted therapy has immensely improved the prognosis of NSCLC, but the overall survival (OS) remains disappointing due to delayed diagnosis and the development of drug resistance, particularly to epidermal growth factor receptor (EGFR) tyrosine kinase inhibitors (TKIs) [3,4]. *EGFR* mutations are one of the most common targetable aberrations that facilitate the progression of malignancies [5]. The advent of EGFR-TKIs has been confirmed to prolong the OS of NSCLC patients. However, almost all patients with *EGFR* mutations invariably acquire EGFR TKI resistance despite the initial encouraging response [6,7]. Although the mechanisms of resistance to EGFR TKIs have been identified as target-dependent mutations and alternative pathway activation [6,7], their underlying mechanisms are still unclear.

Noncoding RNAs, mainly including microRNAs (miRNAs), long noncoding RNAs (lncRNAs) and circular RNAs (circRNAs), show extensive tissue-restricted and cancer-specific expression patterns, which are strongly involved in tumor evolution and the development of drug resistance [8,9]. In the field of EGFR TKI-resistant lung cancer, emerging evidence has verified that dysregulated ncRNAs play an indispensable pathophysiological role by modulating crucial signaling pathways such as PI3K/AKT/mTOR, Ras/Raf/MEK/ERK, JAK/STAT and epithelial-mesenchymal transition (EMT) processes to affect resistance to EGFR TKIs [8,10]. Therefore, ncRNAs show potential as prognostic, predictive and therapeutic biomarkers for EGFR TKI-resistant lung cancer.

This review scientifically summarizes the present knowledge regarding the biogenesis and biological functions of ncRNAs, including the formation of dysregulated ncRNAs in EGFR TKI-resistant lung cancer. In addition, ncRNAs involved in the mechanisms of EGFR TKI resistance and their clinical applications are described in detail. Eventually, the bottlenecks that urgently need to be addressed to enhance the translation of ncRNA basic research to clinical practice are discussed.

## 2. Overview of NcRNAs

For the past few decades, the protein-coding genome has been considered the main focus of medical studies. Human genome sequencing showed that only 1% of the genome had translational function and the remaining regions were initially believed to be “junk” [11,12]. Nevertheless, recent research has indicated that the noncoding portion tends to be transcribed into multiple ncRNAs, which modulate natural pathophysiological processes such as proliferation, invasion and angiogenesis to contribute to the development of diseases, including cancer [13]. On the basis of size and function, ncRNAs can be divided into miRNAs, ncRNAs, circRNAs and so on (Figure 1).

### 2.1. Biogenesis and Role of miRNA

MiRNAs are a category of short noncoding molecules that are approximately 19 to 25 nucleotides in length and primarily function by binding complementary sequences [14]. The biogenesis of miRNA is mediated by RNA polymerase II (Pol II), which consequently forms the primary miRNA (pri-miRNA). Following the original transcription, Drosha, along with the cofactor DGCR8, collaboratively cleaves the pri-miRNA into pre-miRNA. Then, the pre-miRNA is exported to the cytoplasm through the Exportin 5 and Ran-GTP complex and cleaved into a miRNA duplex by TAR RNA binding protein (TRBP) and the RNase III Dicer. Finally, facilitated by the Argonaute (AGO) family of proteins, miRNA duplexes are integrated into the RNA-induced silencing complex (RISC) [15]. Mature miRNAs perform biological functions mainly via the identification of the 3′ untranslated region (UTR) to mediate mRNA degradation or mRNA expression modulation [16]. In this way, miRNAs contribute to the tumorigenesis and evolution of multiple cancers, including lung cancer [17].

### 2.2. Biogenesis and Role of LncRNAs

Unlike miRNAs with short nucleotides, lncRNAs consist of more than 200 nucleotides [18]. Generally, protein-coding genes synthesize mature mRNA by means of 5′ capping, 3′ cleavage and polyadenylation (CPA) and splicing. In certain circumstances, loss of CPA factors, 5′-3′ exoribonuclease 2 (XRN2), or polyadenylation site mutation may lead to prolonged read-through transcript formation. Conversely, premature transcription termination (PTT) induces truncated transcriptional products. The read-through and PTT transcripts are deemed as lncRNAs [19]. Despite distinctly lower expression levels than mRNAs, lncRNAs have a much starker tissue-restricted expression pattern [20] and perform cis-regulatory and trans-acting functions or serve as a molecular scaffold of miRNAs, mRNA, DNA and proteins to participate in physiological regulation and disease pathogenesis [21].

### 2.3. Biogenesis and Role of CircRNA

CircRNAs have emerged as a novel class of circular molecules displaying stronger stability than other ncRNAs due to their covalently closed nature [22]. The biogenesis of circRNA generally involves back-splicing via complementary sequences or RNA-binding proteins (RBPs) and lariat-mediated circularization [23,24]. In accordance with the composition, circRNAs containing exons are considered as exonic circRNAs (ecircRNAs), exerting fundamental biological functions as miRNA sponges and protein scaffolds in the cytoplasm [25,26,27]. CircRNAs consisting only of introns are named intronic circRNAs (ciRNAs), and those with both exons and introns are considered exon-intron circRNAs (EIciRNAs). EIciRNAs and ciRNAs restricted to the nucleus play critical transcriptional regulatory roles [23]. In contrast to other ncRNAs, certain circRNAs containing internal ribosome entry sites (IRESs) or with N6-methyladenosine (m6A) modification possess a translational function, which mediates pathological and physiological regulation via functional proteins or peptides [24,28]. Because of their stable structure and cancer-specific expression pattern, circRNAs can be utilized as promising biomarkers for multiple cancers.

## 3. Dysregulated NcRNAs in EGFR TKI-Resistant Lung Cancer

In research on malignant tumors, dysregulated expression of ncRNAs is widely observed in almost all types of cancers. Nevertheless, the underlying mechanisms remain unclear. Emerging evidence has provided two major hypotheses that partly explain the dysregulation of ncRNA expression in EGFR TKI-resistant lung cancer. If the downregulation of specific ncRNAs occurs during cancer development, one explanation is that the speed of tumor cell proliferation may be faster than that of slow generation ncRNAs, resulting in a dilution effect and significant reduction in ncRNA expression [22,29]. In contrast, individual circRNAs could be upregulated in tumors, potentially resulting from host gene mutation or chromosomal rearrangements. For instance, the oncogene *MYC* mutation observed in multiple cancers transactivates the miR-17–92 cluster and enhances their expression [30]. Analogously, cancer-associated chromosomal rearrangements contribute to the genesis of fusion genes and subsequently result in oncogenic ncRNAs, interfering with the migration and invasion of lung cancer cells [31]. It is commonly assumed that upregulated ncRNAs have an oncologic function. Conversely, the downregulated ncRNAs potentially have a pivotal tumor-suppressive role. Beyond tumorigenesis, recent studies have investigated dysregulated ncRNAs in EGFR TKI-resistant lung cancer and partially elucidated their diverse mechanisms. To date, research has mainly focused on specific ncRNAs involved in crucial signaling pathways downstream of *EGFR* and parallel pathways instead of the mechanisms of ncRNA dysregulation [32,33,34]. Remarkably, a minority of studies have shown that *EGFR* mutations might lead naturally to ncRNAs [35]. MiR-21, for example, a miRNA activated by *EGFR*, has been proven to negatively modulate TNF expression. In this manner, miR-21 in lung cancer treated with EGFR TKIs is inversely upregulated and thus stimulates the TNF/NF-κB pathway [35], which consequently causes EGFR TKI resistance in lung cancer. Nevertheless, the current research is incapable of clearly illustrating the underlying mechanisms relating to dysregulated ncRNAs in EGFR TKI-resistant lung cancer. Further investigation focusing on transcriptional control of ncRNAs will likely provide an in-depth understanding of EGFR TKI resistance.

## 4. Mechanisms of NcRNAs Involved in EGFR TKI Resistance

Acquired drug resistance has long been a major hurdle hindering the therapeutic effect of EGFR TKIs [6,7,8]. Accumulating evidence has indicated that ncRNAs, especially circRNAs, lncRNAs and miRNAs, play a vital role in EGFR TKI-resistant lung cancer via alternative signaling pathway modulation [10,36]. Here, ncRNAs involved in essential signaling pathways downstream and parallel pathways of *EGFR* are discussed in detail (Figure 2).

### 4.1. NcRNAs Involved in the PI3K/AKT/mTOR Signaling Pathway

The PI3K/AKT/mTOR signaling pathway is a critical signaling pathway downstream of *EGFR* that dramatically regulates cell proliferation, metabolism and motility and brings about EGFR TKI resistance [37]. Previous studies have validated that many ncRNAs affect the activation of the PI3K/AKT/mTOR signaling pathway by directly interacting with oncoproteins, modulating regulators and targeting effectors of the PI3K/AKT/mTOR pathway upstream.

The field of EGFR TKI resistance research has identified many ncRNAs interacting with oncoproteins, which may directly relieve the trigger of downstream signaling in EGFR TKI-resistant lung cancer and thereby reduce drug sensitivity. For instance, miR-30a-5p and miR-7 functionally attenuate the activation of PI3K and AKT, respectively, hence eliminating resistance to EGFR TKIs [38,39]. Apart from miRNAs, lncRNAs such as H19 and LINC00152 regulate the phosphorylation of AKT to restore erlotinib resistance [40,41]. Furthermore, dysregulated circRNAs serve as miRNA sponges to inactivate PI3K/AKT proteins [42,43], but their mechanisms remain unclear.

Beyond the direct coaction with oncoproteins, ncRNAs are also engaged in modulating specific factors that contribute to silencing or enhancing the PI3K/AKT/mTOR signaling pathway. Among them, phosphatase and tensin homolog (PTEN), a major antagonist of PI3K, decreases this pathway to a great extent [44]. Previous research has revealed that miR-21 silences the expression of PTEN and reinforces the PI3K/Akt pathway, resulting in acquired resistance to gefitinib [45]. In contrast, circ-PLCD1 enhances PTEN expression by binding to miR-375 and miR-1179 [46]. In addition, other modulators, including ENO1 and AGO1, lead to dysregulation of the PI3K/AKT/mTOR signaling pathway [2,33,47,48], which promotes or restricts resistance to EGFR TKIs.

In addition to interactions with oncoproteins and the regulators of the pathway, many ncRNAs contribute to EGFR TKI resistance in patients by modulating alternative signaling pathways, such as the hepatocyte growth factor (HGF) receptor and insulin-like growth factor 1 receptor (IGF1R). The hepatocyte growth factor (HGF) receptor, also known as MET, has been shown to be activated by miR-205/ERRFI1 and LINC01510. Thus, the PI3K/AKT pathway is enhanced, causing resistance to gefitinib, afatinib, erlotinib and osimertinib [49,50]. In a similar manner, ncRNAs such as miR-223, lncRNA GAS5 and hsa_circ_0005576 regulate IGF1R expression and subsequently influence the resistance of lung cancer cells to EGFR TKIs [32,51,52,53].

### 4.2. NcRNAs Involved in the Ras/Raf/MEK/ERK Signaling Pathway

The Ras/Raf/MEK/ERK signaling pathway, a well-researched oncogenic pathway, is responsible for the tumorigenesis and evolution of multiple malignancies [54] and has also been validated to be an indispensable mechanism of EGFR TKI resistance [55]. As a downstream pathway of *EGFR*, the Ras/Raf/MEK/ERK pathway has been confirmed to be modulated by various ncRNAs in recent years.

The majority of ncRNAs exert their biological function by influencing the activity of oncoproteins or regulatory factors of the pathway. MiR-345, for instance, directly acts on the ERK oncoprotein to eliminate the signal transduction of the Ras/Raf/MEK/ERK pathway, thereby regaining sensitivity to gefitinib [56]. Furthermore, other ncRNAs, such as miR-641, miR-630, lncRNA CASC9 and lncRNA LOC554202, maintain EGFR TKI resistance by regulating the enhancers of the pathway [57,58,59,60]. Intriguingly, LOC554202 sustains both the Ras/Raf/MEK/ERK and PI3K/AKT/mTOR pathways by activating miR-31 [58], indicating its potential as a promising therapeutic target for EGFR TKI-resistant lung cancer. In regard to circRNAs, C190 sponges miR-142-5p to promote EGFR/MAPK/ERK signaling [61], which may contribute to EGFR TKI resistance in lung cancer patients. However, the current evidence remains inadequate to clarify its relationship with drug resistance and experiments regarding certain EGFR TKIs are urgently needed.

The effectors of the Ras/Raf/MEK/ERK pathway also activate signaling and accordingly mediate EGFR TKI resistance. MET is one of the best-known effectors that not only stimulates the PI3K/AKT/mTOR pathway but also facilitates the Ras/Raf/MEK/ERK pathway [62]. By directly targeting MET, miRNAs such as miR-1-3p and miR-206 attenuate gefitinib resistance by inactivating the Ras/Raf/MEK/ERK and PI3K/AKT/mTOR pathways [63]. In contrast, lncRNA LINC01510 promotes MET expression to maintain EGFR TKI resistance [50]. Moreover, emerging research has indicated that circBFAR functionally promotes the MET pathway by sponging miR-34b-5p [64], which can be hypothesized to cause EGFR TKI resistance. However, insufficient experimental validation of EGFR TKI-resistant NSCLC cells or tissues obscures the mechanisms of circRNAs in MET-related drug resistance.

### 4.3. NcRNAs Involved in the JAK/STAT and NF-κB Signaling Pathways

Relying on corresponding ligands and receptors, the JAK/STAT signaling pathway exerts specific biological functions, including limited proliferation, apoptosis and even EGFR TKI resistance [65]. Many ncRNAs are involved in silencing or activating JAK/STAT signaling via interconnection with oncoproteins or upstream regulators. As exemplified by circ-E-Cad and lncRNA UCA1, some ncRNAs directly target the JAK/STAT pathway to mediate EGFR TKI resistance [66,67]. However, the current research is more focused on the upstream modulators of the JAK/STAT signaling pathway. For instance, miRNAs, including miR-19b and miR-206, indirectly modulate the JAK/STAT pathway via critical upstream signal control and consequently affect resistance to EGFR TKI [34,68]. Remarkably, miR-19b interacts with PP2A and BIM to simultaneously promote the phosphorylation of AKT, ERK and STAT [34], indicating its potential as a promising therapeutic target for affecting multiple pathways of EGFR TKI-resistant lung cancer. Similarly, lncRNA LINC01116 attenuates the expression of IFI44, a critical upstream regulator of IFN/STAT1 signaling, to expedite gefitinib resistance [69].

Nuclear factor-κB (NF-κB) has long been known as an inducible transcription factor participating in cellular life activities, especially in malignancies [70,71]. In cancer biology, NF-κB has major functions as a proliferative stimulator and apoptotic suppressor in tumor cells, which also mediates drug resistance [70]. Notably, NF-κB not only independently sustains EGFR TKI resistance but also serves as an effector downstream of the JAK/STAT signaling pathway to enhance EGFR TKI resistance [36]. NcRNAs involved in NF-κB pathway modulation are primarily examined for their ability to target upstream regulators. For example, miR-21 induces the downregulation of TNF mRNA, which may contribute to the suppression of TNF-related NF-κB activation and thus reduce EGFR TKI resistance [35]. Conversely, ciRS-7, one of the most studied circRNAs, recruits miR-7 to activate HOXB13-mediated NF-κB, which might promote EGFR TKI resistance [72]. Nevertheless, insufficient research on ncRNAs involved in the NF-κB pathway makes it difficult to clarify their relationship with EGFR TKI-resistant lung cancer and further study is urgently needed to illustrate their underlying mechanisms.

### 4.4. NcRNAs Involved in EMT

EMT is a common cellular phenomenon that crucially leads to embryogenesis and malignant evolution. During tumorigenesis, EMT induces tumor cells to increase their invasive capability and resistance to therapeutic regimens, including chemotherapy and EGFR TKIs [73]. The EMT signaling pathways are predominantly composed of transforming growth factor-β (TGFβ), WNT and NOTCH pathways [73,74]. In the context of ncRNAs, however, EMT of EGFR TKI-resistant lung cancer is mainly restricted to the TGFβ pathway, NOTCH pathway and EMT-inducing transcription factors (Figure 3).

Regarding TGFβ pathway modulation, lncRNAs such as UCA1 and HCP5 can serve as activators of the TGFβ pathway to sustain EGFR TKI resistance [75,76]. Beyond the TGFβ pathway, lncRNA SNHG15, a crucial molecule downstream of the NOTCH pathway, recruits miR-451 to enhance MDR-1 expression and thus facilitate gefitinib resistance [77]. Moreover, many ncRNAs can reinforce EGFR TKI resistance via EMT-inducing transcription factors. For example, miR-200c could affect the expression of zinc finger E-box binding 1 (ZEB1), a critical transcription factor, to sustain the transition to the mesenchymal cell state and hence induce EGFR TKI resistance [78]. Although the molecular mechanisms of ncRNAs involved in EMT described above have been elucidated, many ncRNAs associated with EGFR TKI-resistant lung cancer are merely correlated with several markers of EMT. Previous research has shown that dysregulation of circRNA CCDC66 correlates with aberrant expression of EMT markers involved in EGFR TKI resistance, such as epithelial cadherin (E-cadherin) and neural cadherin (N-cadherin), hypothesizing that these ncRNAs enhance EGFR TKI resistance via the EMT process [79]. However, due to insufficient evidence, the underlying mechanisms are still unclear.

Since EMT is a reversible process, actively intervening in EMT-related heterotypic signals or de-inducing the expression of EMT-inducing transcription factors can convert mesenchymal cells back to an epithelial state [73], which is known as mesenchymal–epithelial transition (MET). Thus, the MET process may reverse the state of EGFR TKI resistance [80]. Consistent with this hypothesis, miR-625-3p targets AXL to alleviate TGFβ-induced EMT, which consequently contributes to gefitinib resistance [81]. Furthermore, miR-506-3p suppresses SHH signaling to upregulate E-cadherin expression and downregulate vimentin expression to resensitize erlotinib-resistant cells [82]. Likewise, overexpression of the lncRNA HOTAIR was found to induce the MET process by stabilizing E-cadherin, as well as de-inducing vimentin and N-cadherin, resulting in resensitization of EGFR TKI-resistant NSCLC cells and a better prognosis of patients [83], demonstrating its potential as a prognostic biomarker and therapeutic target for EGFR TKI-resistant lung cancer.

### 4.5. NcRNAs Involved in Other Mechanisms

The cooperative interaction between FGF2-fibroblast growth factor receptor (FGFR1) and *EGFR* in activating the PI3K/AKT/mTOR, Ras/Raf/MEK/ERK and JAK/STAT signaling pathways has been proven [84]. Accordingly, FGFR1 is generally regarded as another alternative pathway mediating EGFR TKI resistance [6,7]. In terms of ncRNAs, miR-16 targets MEK1, a downstream mediator of FGFR-1, to attenuate ERK expression and restrain the oncogenic capacity [85], which may subsequently suppress EGFR TKI resistance. Apart from FGFR-1/MEK/ERK pathway modulation, miR-214-3p inhibits FGFR1 expression to inactivate the MAPK/AKT pathway [86]. Conversely, overexpression of lncRNA PVT1 was found to functionally sponge miR-551b to upregulate FGFR1, hence promoting the proliferation and metastasis of NSCLC cells [87]. Therefore, lncRNA PVT1 shows potential as a therapeutic target for EGFR TKI-resistant NSCLC. Nonetheless, current studies of ncRNAs involved in the FGFR1 pathway predominantly focus on general NSCLC cells instead of EGFR TKI-resistant NSCLC cells and further research is urgent to illustrate the mechanisms of FGFR1-inducing EGFR TKI resistance mediated by ncRNAs.

SCLC transformation has always been a rare mechanism of acquired resistance to first-, second- and third-generation EGFR TKIs [6,7]. Multicenter research indicated that approximately 3% to 10% of *EGFR*-mutant NSCLC patients undergo SCLC transformation characterized by TP53 and Rb1 mutations [88]. A recent study utilized TKI-resistant profiles to establish the miRNA regulatory network and select hsa-miR-495-3p, hsa-miR-24-3p, hsa-miR-181a-5p and hsa-miR-125a-3p, which potentially participate in SCLC transformation [89]. This discovery indicates that ncRNAs might shed light on the mechanisms of SCLC transformation and an in-depth study dissecting the expression of ncRNAs at the single-cell level would promote the development of treatment concerning SCLC transformation-related EGFR TKI resistance in the future.

## 5. Clinical Implications of NcRNAs in *EGFR*-Mutant Lung Cancer

Dysregulated ncRNAs are characterized by cancer-specific and tissue-restricted expression patterns. Moreover, accumulating evidence has demonstrated that aberrant expression of individual ncRNAs dramatically correlates with TNM stage, treatment response and even prognosis of EGFR TKI-resistant lung cancer, highlighting their potential as prognostic, predictive and therapeutic biomarkers [33,90,91,92]. Here, the clinical implications of ncRNAs in *EGFR*-mutant lung cancer are listed with examples to offer a useful reference for clinical decision-making (Table 1).

### 5.1. NcRNAs as Prognostic Biomarkers

An increasing number of ncRNAs have been confirmed to have immense potential as prognostic biomarkers to promote medical resource allocation. For ncRNAs conferring better prognosis of *EGFR*-mutant NSCLC, earlier multicenter research on 319 EGFR-TKI-treated patients found that miR-608 rs4919510 and miR-4513 rs2168518 contributed to prolonged progression-free survival (PFS) with hazard ratios (HRs) of 0.63 and 0.46, respectively (*p* < 0.01) [93]. Consistent with the survival results, the miRNAs notably enhanced gefitinib sensitivity in H1299 and PC9 cells in vitro [93]. Analogously, overexpression of lncRNA H19 denotes an extended PFS (*p* = 0.021) in *EGFR*-mutated patients under EGFR-TKI treatment. Mechanistically, H19 contributes to erlotinib sensitivity by hindering AKT phosphorylation, resulting in survival improvement [40]. Beyond PFS prediction, some researchers are dedicated to exploring the relationship between overall survival (OS) and ncRNA expression. Loss of heterozygosity (LOH) of microRNA-128b in *EGFR*-mutant patients resulted in an improved OS of 23.4 versus 10.5 months [94]. In contrast, upregulated hsa_circ_0004015 alleviates the inhibition of miR-1183 to promote gefitinib resistance, which correlates with worse OS (*p* < 0.05), invasion (*p* = 0.031) and TNM stage (*p* = 0.004) [95]. However, although overexpression of lncRNA SOX2-OT is significantly associated with metastasis and a lower response to EGFR-TKI treatment, no significant difference in OS was observed [96].

### 5.2. NcRNAs as Predictive Biomarkers

Through promotion of EGFR-TKI resistance and treatment response prediction, ncRNAs are valuable in noninvasive selection of optimal therapeutic regimens without extra toxicities. Considering the emerging preclinical evidence that ncRNAs regulate the efficacy of EGFR TKIs in vitro, ncRNA expression levels in specimens have been explored in accumulating studies to predict resistance to EGFR TKIs [97,98]. For instance, the expression of lncRNA CCAT1 was found to be upregulated in gefitinib-resistant patients compared with gefitinib-sensitive patients (*p* < 0.001) [97]. Beyond the prediction of EGFR-TKI resistance, upregulated LINC00460 in tumor specimens was correlated with worse PFS and OS (*p* = 0.046 and 0.014, respectively) [98]. Mechanistically, LINC00460 could serve as a miR-149-5p sponge to enhance IL-6 expression and thus facilitate EMT [98]. Remarkably, large amounts of ncRNAs are secreted in the form of exosomes, which makes it possible to noninvasively detect them in plasma or serum. MiR-7, the best-studied miRNA, is positively associated with gefitinib sensitivity in serum exosomes of *EGFR*-mutant LUAD patients (*p* < 0.0001) [99]. In contrast, miR-27a, miR-21 and miR-218 have been verified to be significantly overexpressed in the plasma of EGFR TKI-resistant NSCLC patients compared with sensitive patients (*p* = 0.009, 0.004 and 0.041, respectively) [100]. Interestingly, previous research constructed a united model containing four plasma miRNAs with significantly aberrant expression to achieve an area under the curve (AUC) of 0.869 in predicting EGFR-TKI resistance, which is better than that of any single miRNA [101]. This instructive example indicates that comprehensively utilizing multidimensional information such as ncRNAs and clinical data may attain a promising predictive performance for the treatment response of EGFR TKIs.

Other researchers have also studied the possible correlation between dynamic variations in ncRNAs in the process of TKI treatment to monitor acquired EGFR TKI resistance. Clinical investigation of miR-184 and miR-3913-5p in serum presented AUCs of 0.736 and 0.759, respectively, in acquired osimertinib-resistant NSCLC patients with exon 21 L858R [102]. In addition, the expression of miR-3913-5p was dramatically upregulated in osimertinib-resistant patients with the T790M mutation (*p* = 0.013) [102]. Other than miRNAs, exosomal circRNA_102481 is significantly upregulated in serum after the onset of EGFR TKI resistance (*p* = 0.025), demonstrating its potential as a predictive biomarker of acquired resistance [90]. Notably, lncRNA HOTAIR showed a good performance in predicting both acquired and primary EGFR TKI resistance (*p* = 0.0046 and 0.0097, respectively) [83]. The evidence above supports that HOTAIR has the ability to simultaneously monitor multiple means of EGFR-TKI resistance.

### 5.3. NcRNAs as Therapeutic Targets

Given that many ncRNAs participate in the modulation of EGFR TKI resistance via multiple signaling pathways, ncRNAs potentially serve as therapeutic targets or agents for EGFR TKI-resistant lung cancer [28,33,61,91,103,104,105,106,107,108,109,110,111,112].

For ncRNAs in promoting EGFR TKI resistance, loss-of-function treatment is hypothesized to have promising curative effects. Oncogenic miRNAs can be attenuated by antagomiRs. MiR-147b, for instance, has been verified to repress pseudohypoxia and the TCA cycle pathway to mediate osimertinib resistance. Pretreating osimertinib-resistant cells with antagomiR of miR-147b markedly reversed EGFR TKI resistance, indicating the therapeutic role of miR-147b [104]. With regard to lncRNAs and circRNAs, rational usage of novel techniques, including RNA interference (RNAi) and CRISPR/Cas13 systems, also show promising therapeutic effects. RNAi includes interfering RNAs (siRNAs) and short hairpin RNAs (shRNAs), which are commonly applied to trigger gene silencing and cancer therapy [113]. By this means, overexpressed lncRNA APCDD1L-AS1 and BLACAT1 were knocked down by siRNA and shRNA, which subsequently reinstated the sensitivity to icotinib and afatinib, respectively [108,109]. Nevertheless, major challenges such as off-target effects, unsatisfactory specificity and cytotoxicity hinder their clinical application [113]. To address this issue, researchers have developed CRISPR/Cas13 systems with higher specificity and efficiency than RNAi to precisely edit ncRNAs [22]. A more recent study utilized CRISPR/Cas13a to facilitate the cleavage of circRNA C190 and further restrain the expression of EGFR-MAPK-ERK signaling, which efficiently reversed EGFR TKI resistance [61].

For ncRNAs that induce sensitivity to EGFR TKIs, establishing adenoviral/lentiviral vectors and lipid/polymer nanoparticles might enhance their expression and achieve excellent therapeutic effects. MiR-483-3p has been proven to repress the FAK/Erk pathway by targeting integrin beta3 to weaken EGFR TKI resistance. In EGFR TKI-resistant tumor models, forced miR-483-3p expression via a lentiviral vector observably stimulated gefitinib sensitivity, showing its potential as a therapeutic agent [91]. In the field of lncRNAs, increasing RHPN1-AS1 expression via a lentiviral vector resensitized gefitinib-resistant NSCLC cells in a similar manner. Mechanistically, RHPN1-AS1 alleviates TNFSF12 suppression by sponging miR-299-3p to relieve resistance to EGFR TKIs [110]. Nonetheless, gain-of-function therapy by means of constructing viral vectors inevitably generates substantial additional products that might lead to unpredictable adverse events in patients [22]. Notably, certain ncRNAs exert translational functions to modulate EGFR TKI resistance. CircASK1, for example, encodes the ASK1-272a.a protein to compete with ASK1 and thus attenuate ASK1 phosphorylation mediated by Akt1, which activates gefitinib sensitivity [111]. Therefore, ncRNAs encoding tumor-suppressor proteins may serve as a novel therapeutic regimen for EGFR TKI-resistant lung cancer.

**Table 1 cancers-14-04423-t001:** Main prognostic, predictive and therapeutic biomarkers in *EGFR*-mutant lung cancer.

Biomarker Type	NcRNA	Expression	Cancer Type	Biological Function or Role	Reference
**Prognostic biomarker**	microRNA-128b	↓	NSCLC (tissue)	Suppressing *EGFR* expression; a microRNA-128b LOH confers better OS of *EGFR*-mutant patients (*p* = 0.02)	[94]
	miR-608 miR-4513	↓	LUAD (tissue)	Enhancing gefitinib sensitivity in H1299 and PC9 cells; overexpression of miR-608 and miR-4513 indicates a better PFS (HR = 0.63 and 0.46, respectively) (*p* < 0.01)	[93]
	lncRNA H19	↓	NSCLC (tissue)	Promoting erlotinib resistance by enhancing the AKT phosphorylation; overexpression of lncRNA H19 indicates an extended PFS (*p* = 0.021) in *EGFR*-mutated patients	[40]
	circ_0004015	↑	NSCLC (tissue)	Alleviating the inhibition of miR-1183 to promote gefitinib resistance; overexpression of hsa_circ_0004015 indicates a worse OS (*p* < 0.05)	[95]
**Predictive biomarker**	miR-7	↓	NSCLC (serum)	Promoting gefitinib sensitivity by targeting YAP; upregulation of miR-7 significantly correlates with gefitinib sensitivity (*p* < 0.0001)	[99]
	miR-184 miR-3913-5p	↑	NSCLC (serum)	Promoting osimertinib resistance; patients with *EGFR* exon 21 L858R: AUC = 0.736 (miR-184) and 0.759 (miR-3913-5p)	[102]
	miR-195, miR-122, miR-125, miR-21, miR-25	↑	NSCLC (tissue & plasma)	Promoting gefitinib resistance; AUC = 0.869 (model including these miRNAs)	[101]
	lncRNA CCAT1	↑	NSCLC (tissue)	Promoting gefitinib resistance by sponging miR-218; upregulation of CCAT1 significantly correlates with gefitinib resistance (*p* < 0.001)	[97]
	lncRNA HOTAIR	↓	NSCLC (tissue)	Promoting EGFR-TKI sensitivity by modulating EMT; downregulation of HOTAIR significantly correlates with EGFR TKI resistance (*p* = 0.0046, acquired resistance; *p* = 0.0097, primary resistance)	[83]
	circRNA_102481	↑	NSCLC (serum)	Promoting EGFR TKI resistance via the microRNA-30a-5p/ROR1 axis; upregulation of circRNA_102481 significantly correlates with EGFR TKI resistance (*p* = 0.025)	[90]
**Therapeutic biomarker**	miR-147b	↑	NSCLC (tissue)	Therapy target of miR-147b-related TCA cycle dysfunction	[104]
	miR-150	↓	LUAD (tissue)	Therapy target of the miR-150/NOTCH3/COL1A1 pathway	[103]
	miR-483-3p	↓	NSCLC (cell line)	Therapy target for inhibiting integrin beta3 and thus repressing the FAK/Erk pathway	[91]
	miR-30a-5p	↓	NSCLC (cell line)	Therapy target for inhibiting the PI3K/AKT pathway	[112]
	lncRNA APCDD1L-AS1	↑	LUAD (cell line)	Therapy target for the miR-1322/miR-1972/miR-324-3p-SIRT5 pathway	[108]
	lncRNA BLACAT1	↑	NSCLC (cell line)	Therapy target for regulating the STAT3 signaling pathway	[109]
	lncRNA RHPN1-AS1	↓	NSCLC (tissue)	Therapy target for inhibiting the miR-299-3p/TNFSF12 pathway	[110]
	circRNA C190	↑	NSCLC (tissue)	Therapy target for the EGFR/MAPK/ERK pathway	[61]
	circASK1	↓	LUAD (tissue)	Therapy target for activating the ASK1/JNK/p38 pathway	[111]

The “↑” for upregulation and the “↓” for downregulation. Abbreviations: Area under the curve (AUC); Epidermal growth factor receptor (EGFR); Epithelial-mesenchymal transition (EMT); Hazard ratio (HR); Loss of heterozygosity (LOH); Lung adenocarcinoma (LUAD); Non-small cell lung cancer (NSCLC); Overall survival (OS).

## 6. Conclusions and Future Challenges

Over the last few decades, ncRNAs have been deemed nonfunctional coproducts of protein-coding RNAs, even though they constitute the overwhelming majority of RNAs [11,12]. The advent of RNA-sequencing and analytical technologies has strongly enhanced the exploration of ncRNA biological functions and pathophysiological roles. On this account, ncRNAs are pyramidally utilized to illustrate the tumorigenesis and progression of malignancies, creating the conditions for in-depth research on the mechanisms of EGFR TKI-resistant lung cancer [10,36]. Emerging evidence has also shown that certain ncRNAs play an indispensable role in the regulation of critical signal pathways inducing EGFR TKI resistance, including PI3K/AKT/mTOR, Ras/Raf/MEK/ERK, JAK/STAT and EMT [38,58,61,68,77].

Although research on ncRNAs has strongly increased our understanding of the underlying mechanisms of EGFR TKI resistance, most studies are limited to target-independent pathways. Consequently, the role of ncRNAs in the molecular network of EGFR TKI-resistant mutations remains unclear. In practice, a few ncRNAs derived from drug-resistant-mutated lung cancer cells have been found to facilitate EGFR TKI resistance [99,114]. Therefore, further exploration of the mechanisms of ncRNAs in patients with EGFR TKI-resistant mutations would theoretically contribute to the comprehensive understanding of the development of EGFR TKI resistance. Despite the ambiguity regarding the mechanisms of EGFR TKI resistance, ncRNAs have long been expected to be promising biomarkers and therapeutic targets [40,61,83,94,97]. Nevertheless, no ncRNAs have truly satisfied the clinical demands and have been put into application. Several major hindrances are discussed briefly below.

The dysregulation of ncRNAs has cumulatively been affirmed to have excellent predictive and prognostic performance, but most studies are based on dozens of cases, making the results unreliable [40,90,101,102]. Hence, further investigation with a large sample size and well-designed validation cohorts is prospectively needed to identify biomarkers satisfying clinical needs. Furthermore, repression or enhancement of functional ncRNAs has been used to affect the natural progression of lung cancer and reverse EGFR TKI resistance in vivo and in vitro [61,104,106,107], indicating their therapeutic value. Nevertheless, the inescapable off-target effects and undesirable viral DNA integration in current technologies such as RNAi, CRISPR/Cas systems and adenoviral/lentiviral vectors are worrisome [10,22]. Ultimately, the particularity of ncRNA structure constitutes an impediment for practitioners. The leading concern for miRNAs and lncRNAs is their unstable nature and accordingly, these molecules are easily degraded and difficult to detect [12,115]. With regard to circRNAs, although they possess a covalently closed structure with high stability, their current profiling techniques are complicated and costly [22], leading to an insurmountable obstacle for clinical application. All things considered, the aforesaid challenges need to be overcome to expedite their translation from basic research to clinical practice.

In summary, ncRNAs show substantial translational prospects. A future study adopting a rational trial design and advanced technologies to explore the ncRNAs involved in EGFR TKI-resistant lung cancer would overcome barriers that hinder mechanistic investigation and clinical application.

## Figures and Tables

**Figure 1 cancers-14-04423-f001:**
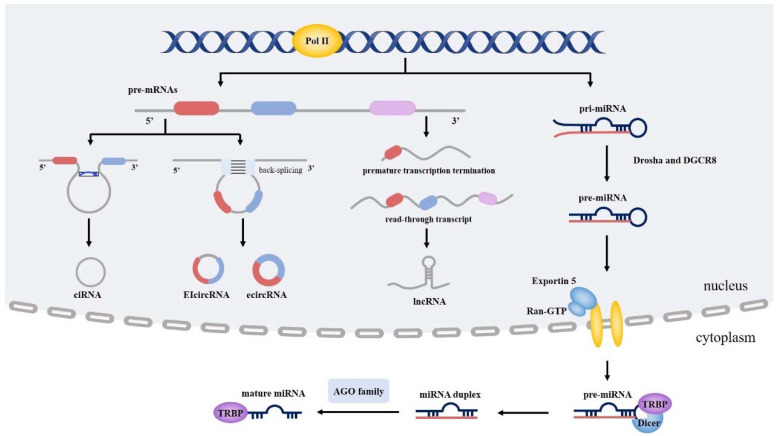
Biogenesis of ncRNAs. Abbreviations: AGO, argonaute; circRNA, circular RNA; ciRNA, intronic circRNA; ecircRNA, exonic circRNA; EIciRNA, exon–intron circRNA; lncRNA, long non-coding RNA; miRNA, microRNA; ncRNA, non-coding RNA; Pol II, RNA polymerase II; TRBP, TAR RNA binding protein.

**Figure 2 cancers-14-04423-f002:**
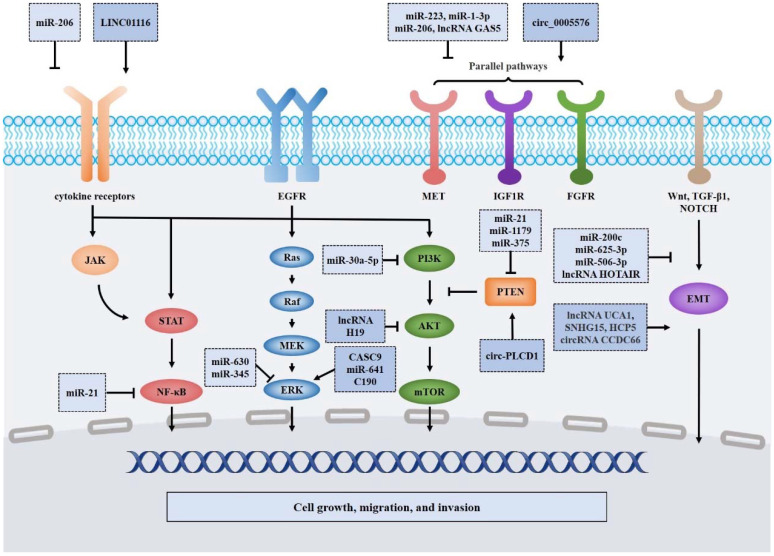
Mechanisms of EGFR-TKI resistance mediated by ncRNAs. Several critical ncRNAs can mediate the EGFR-TKI resistance through modulating the PI3K/AKT/mTOR, Ras/Raf/MEK/ERK, JAK/STAT, NF-κB and EMT pathways. Abbreviations: EGFR, Epidermal growth factor receptor; EMT, epithelial-mesenchymal transition; IGF1R, Insulin-like growth factor 1 receptor; PTEN, phosphatase and tensin homologue; TGF-β, transforming growth factor-β.

**Figure 3 cancers-14-04423-f003:**
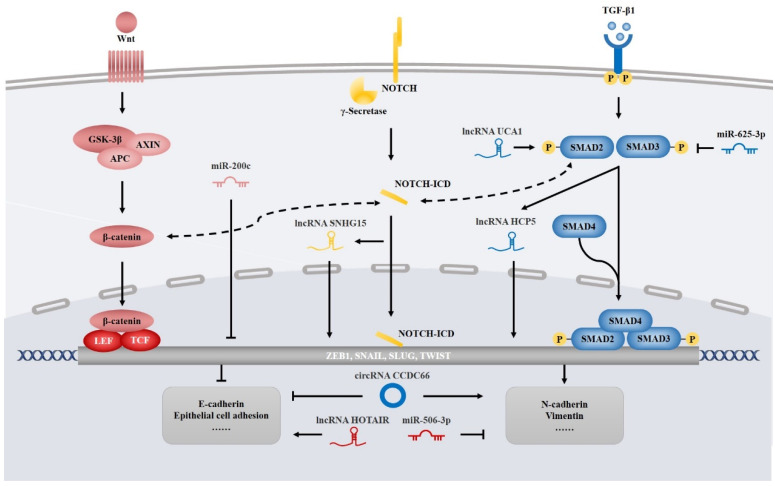
NcRNAs involved in EMT. Dysregulated ncRNAs play an important role in EGFR-TKI resistance via EMT regulation. Abbreviations: APC, adenomatous polyposis coli protein; AXIN, axis inhibition protein; E-cadherin, epithelial cadherin; EMT, epithelial-mesenchymal transition; GSK3β, glycogen synthase kinase-3β; LEF, lymphoid enhancer-binding factor; N-cadherin, neural cadherin; NOTCH-ICD, intracellular domain of the NOTCH receptor; TCF, T cell factor; TGF-β, transforming growth factor-β.

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
