# Peer review of "Emerging Role of Noncoding RNAs in EGFR TKI-Resistant Lung Cancer"

_cancers, 2022, doi:10.3390/cancers14184423_

Round 1

Reviewer 1 Report

The paper titled " Emerging role of non-coding RNAs in EGFR TKI- resistant lung cancer” provides a broad picture of how Tyrosine kinase inhibitor (TKI) therapy becomes challenging with impeding TKI resistance and the potential mechanisms by which non-coding RNAs mediate this TKI resistance. Recently, a growing body of work has explored the impact of non-coding RNAs and their functional roles in diseases including malignancies. This timely review summarizes the known non-coding RNAs connected to TKI resistance and epithelial-mesenchymal transition in lung cancer. In addition, this review also described the challenges to be addressed to promote the clinical applications of these non-coding RNAs in future. This review is comprehensive and informative. I commend the authors for taking a deep dive into the previously reported data on non-doing RNAs and the potential mechanisms of TKI resistance. I find the review valuable to the field. I have only a few minor comments to be addressed.

Here are my comments to authors.

1.    Page 4, line 55: The authors describe circRNAs have translational function in contrast to the other non-coding RNAs. Can the authors add a few sentences on how these circRNAs are translated (internal ribosome entry site (IRES) present in these circRNAs and thereby it gets translated independent of 5’ cap) to give the reader a little more insight into the circRNA translation?

2.    Page 8, lines 343-349: The authors discuss about miR-21 reduces the EGFR TKI resistance while miR-7 promotes EGFR TKI resistance. Can the authors comment on were these studies conducted in different contexts? Does this miR-21 function differently than the most other reported miRNAs that promote TKI resistance?

3.    Typos: Line 225, line 323.

4.    Table 1 needs to be formatted to accommodate all the content.

Reviewer 2 Report

Li. et have provided a nice review on the role of ncRNA in TKI resistance Lung tumors. Although this topic is relatively well-reviewed in the literature, I still support the publication of this review given the ever-evolving nature of underlying science. It is especially important since ncRNA can be a nice biomarker for the early discovery of cancer progression and drug resistance. However, I suggest some revisions before the acceptance to strengthen the data-presentation. My comments are as follows;

1. Biogenesis of ncRNA: I will suggest shortening this part (2.1-2.3) of the review. ncRNA biogenesis has been exclusively reviewed in many good reviews. In this review, it seems a bit off-topic (lung cancer subjected) and also unwarranted.

2. Lack of summary figure: The description on ncRNA in various aspects of lung cancer progression (section 4) is very comprehensively written. However, a summarisation figure for this would be appreciated instead of figures only on two independent events (Fig2 - TKI resistance signaling pathway, and Fig.3- EMT). So summaristic one figure with molecular and biological flow will be appreciated.

3. English: Many a times, the English written is too "emotional" and not scientific. Examples;

 (i)The apocalypse of example is that comprehensively utilizing 507 multi-dimensional information such as ncRNAs and clinical data may attain a promising predictive performance for treatment 509 response of EGFR TKI. What do you mean-apocalypse of example.

(ii)Some other researchers have always also embarked on putative correlation between the dynamic variation of ncRNAs in the processes of TKI treatment to monitor the acquired EGFR TKI 513 resistance

Such examples of improper use of English are widespread throughout the manuscript. Please correct this.

4. For a table on the clinical significance of ncRNA in lung cancer TKI-resistance. I will replace such hard-to-read repetitive tables with a creative figure. But this point is not absolutely necessary.

After these points are addressed, I recommend the publication of this review.
